# Chiral structures of electric polarization vectors quantified by X-ray resonant scattering

Kook Tae Kim[1,15], Margaret R. McCarter[2,15], Vladimir A. Stoica [3,4], Sujit Das[5,14], Christoph Klewe[6], Elizabeth P. Donoway[2], David M. Burn [7], Padraic Shafer [6], Fanny Rodolakis [3], Mauro A. P. Gonçalves [8], Fernando Gómez-Ortiz [9], Jorge Íñiguez [10,11], Pablo García-Fernández [9], Javier Junquera [9], Sandhya Susarla [12], Stephen W. Lovesey[7], Gerrit van der Laan [7], Se Young Park [1], Lane W. Martin [5,13], John W. Freeland [3✉], Ramamoorthy Ramesh [2,5,13✉] & Dong Ryeol Lee [1✉]

Resonant elastic X-ray scattering (REXS) offers a unique tool to investigate solid-state systems providing spatial knowledge from diffraction combined with electronic information through the enhanced absorption process, allowing the probing of magnetic, charge, spin, and orbital degrees of spatial order together with electronic structure. A new promising application of REXS is to elucidate the chiral structure of electrical polarization emergent in a ferroelectric oxide superlattice in which the polarization vectors in the REXS amplitude are implicitly described through an anisotropic tensor corresponding to the quadrupole moment. Here, we present a detailed theoretical framework and analysis to quantitatively analyze the experimental results of Ti $L$-edge REXS of a polar vortex array formed in a $PbTiO_3$/$SrTiO_3$ superlattice. Based on this theoretical framework, REXS for polar chiral structures can become a useful tool similar to x-ray resonant magnetic scattering (XRMS), enabling a comprehensive study of both electric and magnetic REXS on the chiral structures.

[1] Department of Physics, Soongsil University, Seoul 06978, Korea. [2] Department of Physics, University of California, Berkeley, CA 94720, USA. [3] Advanced Photon Source, Argonne National Laboratory, Lemont, IL 60439, USA. [4] Department of Materials Science and Engineering, Pennsylvania State University, Pennsylvania, PA 16802, USA. [5] Department of Materials Science and Engineering, University of California, Berkeley, CA 94720, USA. [6] Advanced Light Source, Lawrence Berkeley National Laboratory, Berkeley, CA 94720, USA. [7] Diamond Light Source, Harwell Science and Innovation Campus, Didcot, Oxfordshire OX11 0DE, UK. [8] Institute of Physics of the Czech Academy of Sciences, Prague, Czech Republic. [9] Departamento de Ciencias de la Tierra y Física de la Materia Condensada, Universidad de Cantabria, Santander, Spain. [10] Materials Research and Technology Department, Luxembourg Institute of Science and Technology (LIST), Avenue des Hauts-Fourneaux 5, Esch-sur-Alzette L-4362, Luxemburg. [11] Department of Physics and Materials Science, University of Luxembourg, 41 Rue du Brill, Belvaux L-4422, Luxembourg. [12] National Center for Electron Microscopy, Molecular Foundry, Lawrence Berkeley National Laboratory, Berkeley, CA 94720, USA. [13] Materials Sciences Division, Lawrence Berkeley National Laboratory, Berkeley, CA 94720, USA. [14]Present address: Department of Material Research Centre, Indian Institute of Science, Bangalore 560012, India. [15]These authors contributed equally: Kook Tae Kim, Margaret R. McCarter. ✉email: freeland@anl.gov; rramesh@berkeley.edu; drlee@ssu.ac.kr

Recently, novel types of polar chiral structures have been discovered in epitaxial ferroelectric $PbTiO_3/SrTiO_3$ superlattice structures. Depending on the number of layer repetitions, number of unit cells in each layer, and type of substrate, various phases such as $a_1/a_2$ ferroelectric domains (two types of $a$-domains with the $a$-axis normal to the growth plane), vortices, and skyrmions have been reported[1–6]. Such chiral polar structures are receiving much attention due to the development of new material properties such as negative capacitance as well as basic scientific interest[7]. In order to understand this newly discovered polar structure in detail, calculations such as phase-field modeling and second-principles calculations were performed, and the results were almost identical to the polarization-vector distribution measured by scanning transmission electron microscopy (STEM)[1,4,8]. Observation using STEM has the advantage of obtaining a real-space image, but since it is a local observation, it is difficult to clearly know whether the observed structure has a long-range order that extends over the entire sample.

In addition, for technological application of polar chiral structures, nondestructive in situ characterization of structural responses to electrical, optical, and mechanical external signals is required[2,9]. To solve this issue, synchrotron X-ray diffraction has been employed. Reciprocal space maps (RSM) using hard X-rays were measured for polar vortex as well as skyrmion structures, and several satellite peaks were observed in the lateral $q_x$ direction[1,2,4]. From this result, it was revealed that the polar chiral structure forms a long-range periodic supercell structure in the lateral direction of the sample. However, this measurement method is insufficient to obtain a comprehensive three-dimensional picture of the polarization texture, and since the measurement is not sensitive to chirality, the three-dimensional structure and the handedness of the vortex is difficult to fully elucidate.

Resonant elastic X-ray scattering (REXS) can provide information about the three-dimensional chiral structure through the interaction between the X-ray polarization vector and electromagnetic multipoles in a sample, especially the magnetization vector and charge quadrupole moment[10]. It is well known that circular dichroism in X-ray resonant magnetic scattering has been observed for magnetic chiral domains[11–17]. Motivated by these studies, REXS was applied to the chiral polar vortices, and X-ray circular dichroism (XCD) was observed in the satellite peak corresponding to the domain period[8]. At first glance, this result can be understood as having a REXS amplitude in which the polarization vector interacts directly with the X-ray polarization vector, like in the case of magnetic scattering. In a recent work by Lovesey and van der Laan[18], it was shown that for the dipole transition of $L$-edge REXS, the parity-odd polarization vector cannot directly replace the parity-even magnetization vector. Instead, the origin of nonmagnetic XCD lies in the periodic modulation of anisotropic tensors (ATs) due to the parity-even charge quadrupole moment, which are well known to give rise to Templeton and Templeton scattering (TTS)[19,20]. However, this study obtained the AT value for the polarization vector in an arbitrary direction by rotating the AT of the basis atom, without experimental verification whether this assumption was correct. This assumption is valid in TTS because, in the case of a single crystal, the site symmetries of the resonant atoms are all identical and their surroundings can always be expressed as rotational operations. On the other hand, in the case of a polar chiral structure formed in a thin film, since the surrounding environment of resonant atoms at different positions changes gradually, the site symmetry is very low, making it difficult to describe it as a rotational operation. In addition, it is difficult to quantitatively analyze experimental data because the method for determining the AT-component value of the basis atom is not yet known.

Here, we present a detailed theoretical framework to quantitatively analyze the X-ray resonant scattering intensity from a chiral structure of electric-polarization vectors. First, we present a method to obtain the component of the resonant-scattering amplitude AT for a resonant atom with an electric-polarization vector along an arbitrary direction. In particular, we verify whether the approximation obtained by rotating the AT of the basis atom is valid by comparing it with a first-principles calculation, and also present a method to obtain the basis AT. In addition, for a realistic model that can be compared with the experimental results, the REXS theory is extended by introducing a coherence length considering random fluctuations of the polarization vectors and a vertical correlation length between bilayers in the superlattice. Next, based on this theoretical framework, we present an experimental analysis of the Ti $L$-edge REXS for a polar vortex array formed in a $PbTiO_3/SrTiO_3$ superlattice. To this end, we present a phenomenological model for a polar vortex array to perform REXS calculations for various three-dimensional chiral structures. From these calculations, we found that the REXS changes sensitively when the core positions of the vortex pairs are staggered (i.e., alternately displaced along the vertical direction). The model that best explains the experimental results is very similar to the STEM results and second-principles calculations, showing a consistent picture of the polar vortex structure.

## Results

**Anisotropic tensor in X-ray resonant-scattering amplitude.** When an X-ray with resonance energy $E$ is incident on an atom and resonant scattering occurs, the ground state $\zeta_\nu$ with energy $E_\nu$ is excited to the state $\psi_\eta$ with energy $E_\eta$, and subsequently deexcites back to the initial state. It undergoes an elastic scattering process that emits X-ray photons with energy $E$. In the electric-dipole transition approximation, this resonant-scattering amplitude can be expressed as[21]

$$f_{\epsilon'\epsilon} = \sum_\eta \frac{\langle \zeta_\nu | \hat{\boldsymbol{\epsilon}}' \cdot \mathbf{R} | \psi_\eta \rangle \langle \psi_\eta | \hat{\boldsymbol{\epsilon}} \cdot \mathbf{R} | \zeta_\nu \rangle}{E - (E_\eta - E_\nu) + i\Gamma} \quad (1)$$

where $\hat{\boldsymbol{\epsilon}}$ and $\hat{\boldsymbol{\epsilon}}'$ are the unit polarization vectors of the incident and scattered X-ray photons, $\Gamma$ is the lifetime-broadening width of the core hole, and $\mathbf{R}$ is the position operator. This scattering factor can be expressed differently depending on the site symmetry around the atom. When considering the magnetic moment $\hat{\mathbf{M}}$ of the resonant atom, it is well known that it can be written as $f_{\epsilon'\epsilon} = (\hat{\boldsymbol{\epsilon}}' \cdot \hat{\boldsymbol{\epsilon}}) f_0(E) + (\hat{\boldsymbol{\epsilon}}' \times \hat{\boldsymbol{\epsilon}}) \cdot \hat{\mathbf{M}} f_1(E) + (\hat{\boldsymbol{\epsilon}}' \cdot \hat{\mathbf{M}})(\hat{\boldsymbol{\epsilon}} \cdot \hat{\mathbf{M}}) f_2(E)$ assuming cylindrical symmetry[10,22]. On the other hand, in the case where the site symmetry of the atom is determined by the crystal structure like a perovskite, it can be expressed in the form of an anisotropic tensor (AT) $\mathbf{T}$, which is a $3 \times 3$ matrix, and can be written

$$f_{\epsilon'\epsilon} = \hat{\boldsymbol{\epsilon}}' \cdot \mathbf{T} \cdot \hat{\boldsymbol{\epsilon}} = \sum_{ij} \epsilon'_i \epsilon_j T_{ij} \quad (2)$$

$$T_{ij} = \sum_\eta \frac{\langle \zeta_\nu | R_i | \psi_\eta \rangle \langle \psi_\eta | R_j | \zeta_\nu \rangle}{E - (E_\eta - E_\nu) + i\Gamma} \quad (3)$$

Such AT scattering is also known as Templeton and Templeton scattering, where a forbidden Bragg peak of a single crystal is observed in REXS[19,20,23–25]. The procedure for calculating the AT-scattering amplitude is as follows. (1) A resonant atom at a special position such as a zero point or Wyckoff site can be preferentially chosen as the basis atom, so that the AT can be easily determined using the site symmetry. (2) ATs of the remaining resonant atoms can be obtained using the rotation of

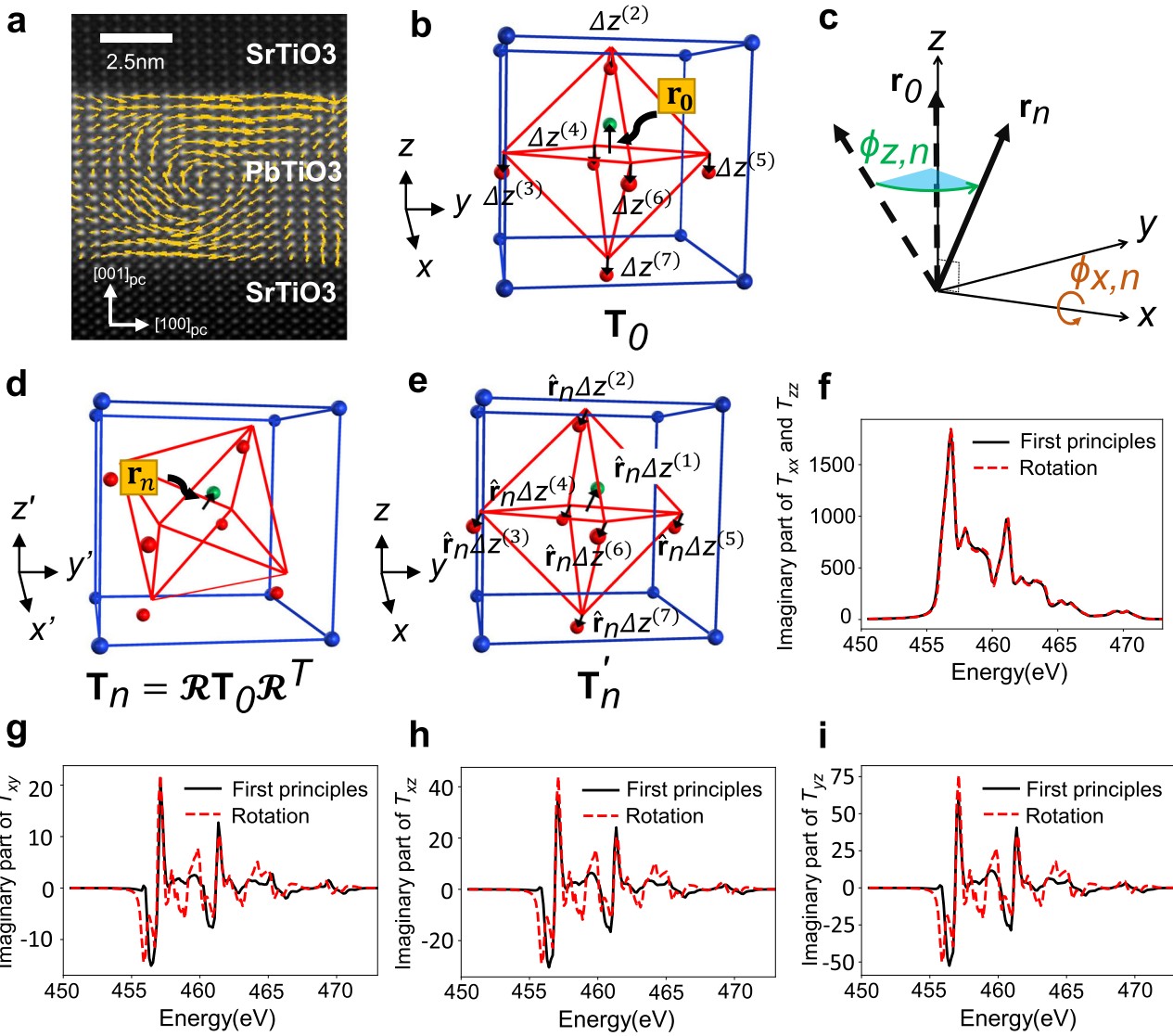

**Fig. 1 X-ray resonant-scattering anisotropic tensor(AT) with electric polarization vector pointing along the arbitrary direction. a** STEM image of a polar vortex in $(PbTiO_3)_n/(SrTiO_3)_n$ ($n = 16$ unit cells) superlattice on $DyScO_3$ $(0\ 0\ 1)_{pc}$ substrate, where $pc$ stands for the pseudocubic direction. **b** Schematic of $ABO_3$ perovskite structure for basis B-site ion. B-site ions (green) are surrounded by octahedral oxygen (red) and A-site ions (blue). The basis B-site ion's displacement vector from the unit-cell center is $r_0$, and $T_0$ is the basis ion's AT. $\Delta z^{(m)}$ is the polar distortion of the $m$th atom having polarization along the $z$-direction and is obtained by relaxing the atomic position of tetragonal $PbTiO_3$. **c** B-site ion and octahedral oxygens rotated with respect to the center of the unit cell. The B-site ion's displacement vector $r_n$ from the unit-cell center position is obtained by rotating the basis ion's displacement vector $r_0$, and the AT $T_n$ is obtained by rotating the basis AT ion's $T_0$. $\mathcal{R}$ is the rotation matrix. **d** Displacement vector $r_n$ obtained by rotating $r_0$ about the $x$- and $z$-axis by the angle $\phi_{n,x}$ and $\phi_{n,z}$, respectively. **e** Polar distortion along the arbitrary direction $\hat{r}_n$ defined as $\hat{r}_n \Delta z^{(m)}$ for the $m$th constituent atom. **f** First-principles calculation of $T_0$ with the polar distortion $\Delta z^{(m)}$ **g-i** Off-diagonal components of the ATs $T'_n$ and $T_n$ obtained by the first-principles calculation of polar distortion $\hat{r}_n$ and the coordinate transformation, respectively.

the AT of the basis atom. (3) The distance between the atoms appears in the phase factor.

**Anisotropic tensor of the basis resonant ion in perovskite crystals.** In the case of AT scattering of a single crystal, the AT components of the basis atom can be simply expressed by using the point-group symmetry of the environment surrounding the resonant atom[20]. In this case, the number of independent components of the AT is reduced by the crystal symmetry. For the polar vortices, the polarization forms a laterally periodic array of pairs of vortices with alternating clockwise and counterclockwise rotation directions, one of which is typically shown in Fig. 1a. Because of the rotating electric polarization of neighboring atoms,

each resonant ion has a low point-group symmetry, so the AT components can no longer be simplified by the crystal symmetry[18]. However, we can make some assumptions that simplify the calculation of the site-dependent AT. First, in the ferroelectric $ABO_3$ perovskite structure, the polarization vector **P** is determined by the displacement vector **r** of $Ti^{4+}$ with respect to the unit-cell center (Fig. 1a)[5,26]. In this case, the site symmetry of the crystal field felt by the resonant $Ti^{4+}$ will be dominantly determined by its relative position to the nearest oxygen ions.

Because the polarization vector rotates continuously throughout the polar vortices, it is useful to choose a high-symmetry direction for the basis ion. For this, we choose the basis ion to have $P_0$ polarized in the $\pm\hat{c}$ axis direction, which has tetragonal symmetry. For this choice of basis ion, $Ti^{4+}$ has a fourfold

rotational symmetry about the $\hat{c}$ axis and mirror symmetry about the $\hat{a}$ and $(\hat{a} + \hat{b})$ axes. Accordingly, the AT $\mathbf{T}_0$ of the basis ion has a simple form

$$\mathbf{T}_0 = \begin{pmatrix} T_{xx} & 0 & 0 \\ 0 & T_{xx} & 0 \\ 0 & 0 & T_{zz} \end{pmatrix} \tag{4}$$

The components of $\mathbf{T}_0$ were calculated using X-ray linear dichroism of a reference sample, as will be discussed below.

**Anisotropic tensor of the resonant atom with polarization pointing in an arbitrary direction.** The remaining $Ti^{4+}$ atoms forming the polar vortex array have displacement vectors $\mathbf{r}_n = \mathcal{R}(\phi_{n,x}, \phi_{n,z})\mathbf{r}_0$ obtained by rotating the basis vector $\mathbf{r}_0$ about the $x$- and $z$-axis by the angle $\phi_{n,x}$ and $\phi_{n,z}$, respectively (Fig. 1b–d), where $\mathcal{R}$ is the rotation matrix. We approximate the rotated AT by rotating the crystal field of the $Ti^{4+}$ basis ion in conjunction with the polarization rotation. Therefore, the AT $\mathbf{T}_n$ for a $Ti^{4+}$ ion with $\mathbf{r}_n$ of the rotation angle $\phi_{n,x}$ and $\phi_{n,z}$ can be expressed using the rotation operation[18]

$$\mathbf{T}_n = \mathcal{R}\left(\phi_{n,x}, \phi_{n,z}\right) \mathbf{T}_0 \mathcal{R}\left(\phi_{n,x}, \phi_{n,z}\right)^T \tag{5}$$

where $\mathcal{R}^T$ denotes a transpose of $\mathcal{R}$. The transformation is based on the assumption that the $T_n$ of the $Ti^{4+}$ ion with polarization pointing along an arbitrary direction can be approximated by changing the local coordinates of $\mathbf{T}_0$ having polarization along the $z$-axis (Fig. 1d). To check whether the assumption is valid, we calculate the anisotropic tensor of $PbTiO_3$ using Eq. (3) by evaluating the band energies and matrix elements between core and valence states using first-principles density-functional theory[27]. We first obtain the $\mathbf{T}_0$ with the polar distortion along the $z$-axis (Fig. 1b, f) and $\mathbf{T}'_n$ with the polar distortion along the arbitrary direction $\hat{\mathbf{r}}_n(= \mathbf{r}_n/|\mathbf{r}_n|)$ (Fig. 1e) defined as

$$\Delta \mathbf{r}_n^{(m)} = \hat{\mathbf{r}}_n \Delta z^{(m)}, \tag{6}$$

where the $\Delta \mathbf{r}_n^{(m)}$ is the polar distortion of the $m$th constituent atom with polarization along the direction $\hat{\mathbf{r}}_n$ and $\Delta z^{(m)}$ is the polar distortion of the $m$th atom having polarization along the $z$-direction (Fig. 1b). The polar distortion $\Delta z^{(m)}$ is obtained by relaxing the atomic position of tetragonal $PbTiO_3$. We next compare the $\mathbf{T}'_n$ with $\mathbf{T}_n$ obtained by Eq. (5) as presented in Fig. 1f–i, showing the case of $\hat{\mathbf{r}}_n$ pointing in the direction $(\phi_x, \phi_z) = (30°, 150°)$. All the components of anisotropic tensors obtained by the coordinate transformation are in good agreement with those having distortion in $\hat{\mathbf{r}}_n$ and the same holds for other arbitrary directions, supporting the validity of our approximation.

**Structure factor of polar vortex array in oxide superlattice.** The structure factor $F_{\epsilon'\epsilon}$ for polar vortices forming a laterally periodic array can be written

$$F_{\epsilon'\epsilon} = \hat{\boldsymbol{\epsilon}}' \cdot \boldsymbol{\Psi} \cdot \hat{\boldsymbol{\epsilon}} \tag{7}$$

$$\boldsymbol{\Psi} = \frac{1}{N} \sum_{n=0}^{N-1} e^{-i\mathbf{q}\cdot\mathbf{r}_{ion,n}} \mathbf{T}_n \tag{8}$$

where $N$ is the total number of crystal unit cells constituting the supercell, which is the repeating unit of the periodic array, $\mathbf{q}$ is the X-ray wave-vector transfer, and $\mathbf{r}_{ion,n}$ is the position vector of the $n$th resonant atom. Using the center position $\mathbf{L}_n$ of the unit cell and the displacement vector $\mathbf{r}_n$ of the B-site atom, $\mathbf{r}_{ion,n} = \mathbf{L}_n + \mathbf{r}_n$ (Fig. 1d). The tensor $\boldsymbol{\Psi}$ is the result of summing the ATs of all resonant atoms in the polar vortex supercell considering their phase difference, resulting in a $3 \times 3$ matrix, which can be

referred to as the supercell structure factor of the polar vortex array.

To calculate the total structure factor for resonant scattering, it is necessary to include the contribution of this multilayer structure. In the $PbTiO_3/SrTiO_3$ superlattice, since the polarization vector is negligible in the $SrTiO_3$ layers, the AT components are negligible. Therefore, it is only necessary to consider the ATs and the polar vortex supercell structure factor in the $PbTiO_3$ layers. If the thickness of the repeated $PbTiO_3/SrTiO_3$ bilayer is $\Lambda$ and the total number of bilayers is $N$, the structure factor of the entire superlattice can be calculated as

$$F_{sup,\epsilon'\epsilon} = F_{0,\epsilon'\epsilon} + F_{1,\epsilon'\epsilon}e^{i\tilde{q}_z\Lambda} + F_{2,\epsilon'\epsilon}e^{i\tilde{q}_z2\Lambda} + \cdots + F_{N-1,\epsilon'\epsilon}e^{i\tilde{q}_z(N-1)\Lambda} \tag{9}$$

where $F_k$ represents the structure factor of the $k$th super cell in the $z$-axis direction. To include the absorption effect, a complex variable $\tilde{q}_z = (4\pi/\lambda)\sqrt{n_{eff}^2 - \cos^2\theta_{in}}$ ($\theta_{in}$ is the incident angle and $\lambda$ is the X-ray wavelength) is used. The imaginary part of the x-ray refractive index $n_{eff}$ corresponds to the absorption effect. In the case of $SrTiO_3/PbTiO_3$ superlattices, the effective index $n_{eff} = (n_{STO}t_{STO} + n_{PTO}t_{PTO})/(t_{STO} + t_{PTO})$, where $n_{STO(PTO)}$ and $t_{STO(PTO)}$ are the refractive index and the layer thickness of the $SrTiO_3(PbTiO_3)$ layer, respectively. If identical polar chiral structures are repeated, all $F_k$ are the same. However, STEM or phase-field simulations show that the vortex structures formed in different $PbTiO_3$ layers in a superlattice are not exactly the same.

To obtain the reflected intensity, the absolute square of the structure factor in Eq. (9) should be calculated. Therefore, the cross-terms $F_k^* F_{k'}$ between the supercell structure factors of different layers should be considered in the reflected intensity. If the vertical correlation length $\xi_v$ between different $PbTiO_3$ layers is introduced to consider the imperfect stacking, then the cross-term $F_k^* F_{k'}$ can be expressed as $|F|^2 e^{-|k-k'|/\xi_v}$. Finally, the absolute square of the final structure factor can be written as

$$|F_{sup,\epsilon'\epsilon}|^2 = |F_{\epsilon'\epsilon}|^2 \sum_{k=0}^{N-1}$$
$$\left( e^{-Im[\tilde{q}_z]\Lambda k} + \sum_{k'=k+1}^{N-1} 2\cos(Re[\tilde{q}_z]\Lambda(k'-k))e^{-\frac{|k-k'|}{\xi_v}}e^{-Im[\tilde{q}_z]\Lambda(k+k')} \right) \tag{10}$$

The final reflected intensity can be calculated using Eqs. (7) and (10). Since the term in the parenthesis in Eq. (10) corresponds to the actual superlattice structure factor and is independent of the X-ray polarization vectors $\epsilon'$ and $\epsilon$, it is sufficient to consider only the polar vortex supercell structure factor $F_{\epsilon'\epsilon}$ of one layer of $PbTiO_3$.

Now, the reflected intensity considering the X-ray polarization can be expressed as[28], neglecting skew-linear polarization

$$I = \frac{1}{2}(1 + P_L)\left(|F_{\sigma'\sigma}|^2 + |F_{\pi'\sigma}|^2\right)$$
$$+ \frac{1}{2}(1 - P_L)\left(|F_{\pi'\pi}|^2 + |F_{\sigma'\pi}|^2\right) + P_C Im(F_{\sigma'\pi}^* F_{\sigma'\sigma} + F_{\pi'\pi}^* F_{\pi'\sigma}) \tag{11}$$

where $\sigma/\sigma'$ and $\pi/\pi'$ are the polarization vectors of incident/scattered X-rays polarized perpendicular to and parallel to the scattering plane, respectively. $P_L$ and $P_C$ denote the degrees of linear and circular polarization of incident X-rays, respectively. Here we explore the case where the incident X-ray is circularly polarized, so that $P_L = 0$ and $P_C = \pm 1$. When the reflected intensity for right- and left-circular polarization is $I_R$ and $I_L$, respectively, the asymmetry ratio (AR) corresponding to the XCD effect is

$$AR = \frac{I_R - I_L}{I_R + I_L} \tag{12}$$

One of the advantages of the AR is that the interface roughness effect and the superlattice structure factor, which are not related

to the polar vortex, are canceled out in the denominator and numerator, so that it solely contains the contribution of the polar vortex supercell structure factor.

Since the vortex pairs are not perfectly repeated throughout the layer, we need to consider how random disorder will alter the scattering. Considering isotropic small fluctuations $\langle \delta r^2 \rangle$ of electric-polarization vectors and the covariance between the polarization vectors of unit cell $n$ and $n'$ expressed as $\langle \delta r^2 \rangle \exp(-\frac{|n-n'|}{\xi_c})$, the average for structure factors in Eq. (11) is expressed as

$$\langle F^*_{\alpha\beta} F_{\gamma\omega} \rangle = \frac{e^{-q^2\langle \delta r^2 \rangle}}{N^2} \sum_{n,n'} e^{i\mathbf{q}\cdot((\mathbf{L_n}-\mathbf{L_{n'}})+(\langle\mathbf{r_n}\rangle-\langle\mathbf{r_{n'}}\rangle))}(\boldsymbol{\alpha}\cdot \mathbf{T^*_n}\cdot\boldsymbol{\beta})(\boldsymbol{\gamma}\ \mathbf{T_{n'}}\cdot\boldsymbol{\omega})e^{q^2\langle \delta r^2 \rangle e^{-\frac{|n-n'|}{\xi_c}}},$$

(13)

where $\alpha$, $\beta$, $\gamma$, and $\omega$ all denote X-ray polarization, <…> denotes a statistical average, and $\xi_c$ is the coherence length. (see Supplementary Note 1 for details). As the coherence length $\xi_c$ decreases, the contribution of cross-terms with $n\neq n'$ decreases, so in the $q_z$-dependent AR curve, which will be described later, as $q_z$ increases, the amplitude of oscillation decreases and broadens.

**Experimentals.** Superlattices of $(PbTiO_3)_n/(SrTiO_3)_n$ ($n = 16$ unit cells) were grown epitaxially on $DyScO_3$ $(001)_{pc}$ substrates (where pc stands for the pseudocubic direction) using pulsed laser deposition. The ferroelectric polarization in the $PbTiO_3$ layers forms in pairs of clockwise- and counterclockwise-rotating vortices, as reported in previous studies[1–3,5]. The polar vortices form

an array along the $[100]_{pc}$ direction, and extend like tubes along the $[010]_{pc}$ direction. In nonresonant hard X-ray diffraction reciprocal space maps (Fig. 2a), there are superlattice peaks along the truncation rod (the $[001]_{pc}$ direction), as well as lateral satellite peaks due to periodic distortions of the lattice. Comparing the vertical line profiles crossing the $DyScO_3$ $(004)_{pc}$ peak and the lateral satellite peaks, there is a clear difference (Fig. 2b). In the former, the total thickness fringe of the superlattice is clearly visible, whereas in the latter, only broad superlattice peaks are observed. This indicates that the vertical correlation length does not reach the total thickness because the polar vortex arrays of the $PbTiO_3$ layers separated by the $SrTiO_3$ layers are structurally different from each other or do not align coherently in the vertical direction.

To probe the three-dimensional chiral structure, REXS experiments at Ti $L_{2,3}$ edges in the soft X-ray regime were performed. To obtain a circular dichroism measurement sensitive to the chiral structure of a polar vortex array, REXS intensities were measured for left- and right-circularly polarized beams at $q_x = 0.057$ Å$^{-1}$, which is the lateral satellite-peak location corresponding to ~11 nm spacing of the polar vortex pair. To elucidate the three-dimensional structure of a polar vortex array in which resonant ions have different polarization vectors in the lateral direction as well as in the depth direction, it is necessary to measure scattering in reciprocal space in three dimensions as well. To accomplish this, a $q_z$ rod scan is performed while maintaining the nonzero $q_x$ component corresponding to the lateral periodicity of the polar vortex array. In this scattering

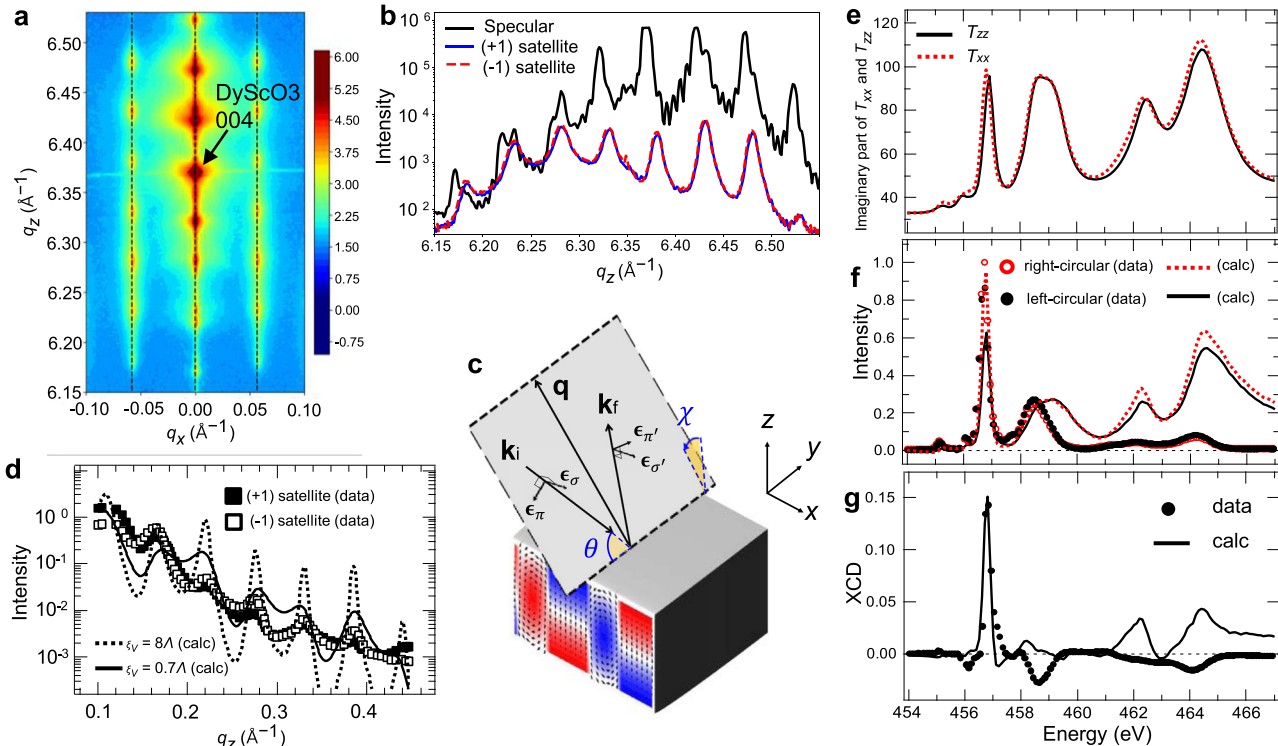

**Fig. 2 Hard X-ray nonresonant and soft X-ray resonant scattering from a polar vortex array. a** Reciprocal space map of $(PbTiO_3)_n/(SrTiO_3)_n$ ($n = 16$ unit cells) superlattice around the $(0\ 0\ 4)_{PC}$ diffraction peak of the $DyScO_3$ substrate using nonresonant hard X-rays. **b** Vertical line profiles corresponding to the truncation rod across the $DyScO_3$ $(0\ 0\ 4)_{PC}$ peak and the $q_z$ rod scan across the lateral satellite peaks due to the periodicity of the polar vortex array formed in the $PbTiO_3$ layer. **c** Scattering geometry. Angles $\theta$ and $\chi$ represent the incident angle and the sample tilt angle generating the lateral $q_x$ component, respectively. **d** REXS intensities of $q_z$ rod scans measured at the first-order satellites of opposite signs. The calculations show the effect of the vertical correlation length $\xi_v$ between $PbTiO_3$ layers. **e** AT components of the basis $Ti^{4+}$ ion obtained from XAS and XLD of a monodomain $PbZr_{0.2}Ti_{0.8}O_3$ thin film measured by Arenholz et al.[33]. **f, g** Energy dependence of REXS intensities with right- and left-circularly polarized X-rays and their difference corresponding to X-ray circular dichroism. Intensities were measured at the lateral satellite-peak position ($q_x = 0.057$Å$^{-1}$, $q_z = 0.164$ Å$^{-1}$). The calculations were obtained using the AT components in **e**.

geometry, as shown in Fig. 2c, the lateral $q_x$ component is obtained by tilting the scattering plane by an angle $\chi$ with respect to the surface normal.

Figure 2d shows the REXS intensities of $q_z$ rod scans measured at the first-order satellites of opposite signs. In order to maximize the XCD effect, the X-ray energy was tuned to the $L_3$ $t_{2g}$ energy (around 457 eV) of the $Ti^{4+}$ atom, which yields a strong absorption effect. In this case, the penetration depth at which the X-ray intensity decreases by 1/e in the X-ray absorption spectroscopy (XAS) measurement might be only one $PbTiO_3$ layer close to the surface. However, in the case of soft X-ray scattering, the effective penetration depth can be much deeper because of the interference effect between the X-rays reflected from the interfaces. From the electric-field intensity distributions as a function of incident angle and depth obtained for nonresonant (450 eV) and resonant (456.8 eV) energies using dynamical calculations[29], it can be seen that total thickness fringes are also seen in specular reflectivity even at the resonant energy when $q_z$ is larger than 0.2 Å$^{-1}$.(see Supplementary Note 2 and Supplementary Fig. 1) For dynamical calculations of REXS intensities in a nonspecular $q_z$ rod scan, the distorted-wave Born approximation (DWBA) developed for X-ray resonant magnetic scattering or grazing-incidence small-angle X-ray scattering can be applied[30–32]. On the other hand, the $q_z$ rod-scan intensity calculated using Eqs. (9)–(10) considering the absorption effect shows that the effective penetration depth is shorter than the total thickness because the total thickness fringe is not clearly visible even when the vertical correlation length $\xi_v$ is the same as the total thickness of the superlattice (Fig. 2d). However, in order to explain the amplitude of oscillation of the experimental data, it can be seen that the vertical correlation length should not exceed one bilayer (Fig. 2d).

**Basis of AT-determination measurement**. To calculate the actual structure factor, it is necessary to know the values of $T_0$. The AT has a real part and imaginary part by the denominator of Eq. (3). Experimental quantification of the AT comes from the imaginary part obtained by measuring the polarization-dependent X-ray absorption spectrum (XAS) of the sample, which is then normalized to the nonresonant scattering factors for $Ti^{4+}$. The real part is then obtained through the well-known Kramers–Kronig transformation of the properly normalized data. However, applying this method directly to the polar vortex structure has the following problems. Since XAS is measured for all resonant atoms with polarization vectors in different directions, the AT of the basis atom cannot be separated. Instead, we opted to use the measured X-ray linear-dichroism (XLD) data for a monodomain $PbZr_{0.2}Ti_{0.8}O_3$ (PZT) thin film with the environment most similar to the basis $Ti^{4+}$ atom[33]. Since the $Ti^{4+}$ atom of the PZT thin film polarized along the crystal $\hat{c}$ axis has 4 mm site symmetry, the AT symmetry is the most similar to $T_0$ in the polar vortex array. Each component of the basis atom's AT $T_0$ was calculated using XAS and XLD of the PZT thin film measured by Arenholz et al.[33] $T_{zz}$ can be obtained using XAS data measured for X-ray polarization parallel to the sample surface normal, and $T_{xx}$ can be obtained using XAS data measured when perpendicular to the surface normal (Fig. 2e).

Figure 2f, g shows the REXS intensity and XCD, respectively, calculated using the XAS spectrum. Comparing the calculation results with the experimental data shows that the XCD is maximum at the $L_3$ $t_{2g}$ energy (457.8 eV) of $Ti^{4+}$ atoms, and the magnitude of the XCD at this energy is also quantitatively consistent with the experimental result. On the other hand, at higher energies, the calculated values of both XCD and scattering intensity do not explain the experimental data well. This is

presumed to be because the XAS or XLD used in the calculation is different from the actual component of $T_0$. Nevertheless, since the XCD experiment of $q_z$ rod scan, which will be discussed below, was performed at the maximum XCD energy corresponding to the $L_3$ $t_{2g}$ energy, it is expected that the experimental data can be sufficiently explained by the AT $T_0$ component of the basis atom obtained here near this energy.

**Phenomenological model of polar vortex array**. For calculation of the structure factor of the polar vortex, it is necessary to determine the distribution of the polarization vector $P_n$ inside the supercell. The distribution of $P_n$ can be obtained by using information directly observed from high-resolution STEM images, or through calculations such as phase-field modeling or second-principles calculations[5,8,34,35]. Phase-field modeling and second-principles calculations are very practical because they are performed based on a physical model, but each has limitations on the level of complexity or details available about the lattice structure. Instead, we decided to use a phenomenological model based on those theoretical results that can easily generate a vortex texture by a few structural parameters while having a polarization-vector distribution similar to the second-principles calculations[8].

This phenomenological polar vortex array model mainly consists of a domain and a domain wall. First, the polarization vectors $P_n$ of the domain-wall region corresponding to regions (I) and (III) in Fig. 3a form a vortex and can be expressed using double helices. One is a helix of polarization vectors rotating in the $y$–$z$ plane along the $x$-axis as shown in Fig. 3b, and the other is a helix rotating in the $x$–$y$ plane along the $z$-axis as shown in Fig. 3c. On the other hand, the domain regions corresponding to regions (II) and (IV) in Fig. 3a consist only of the helix (Fig. 3d) of the polarization vector rotating in the $y$–$z$ plane along the $z$-axis. In both regions, since the polarization vector rotates by $\phi_x$ about the $x$-axis in the $y$–$z$ plane or $\phi_z$ around the $z$-axis in the $x$–$y$ plane, the direction of the polarization vector of each unit cell can be determined by both $\phi_x$ and $\phi_z$.

If the domain-wall region [region (I) in Fig. 3a] consists of ($N_w + 1$) and ($N_z + 1$) number of unit cells along the $x$- and $z$-axis directions, respectively, the rotation angle of the polarization vector of the ($n_x$, $n_z$)-th unit cell can be given as $\phi_x = \eta_x \pi n_x / N_w$ and $\phi_z = \eta_{z,w} \pi(n_z/N_z - 1)$, respectively (Fig. 3e). Here, $\eta_x$ and $\eta_{z,w}$ correspond to the rotation direction of each helix, and are +1 for the right-hand screw direction and −1 for the left-hand screw direction. As shown in Fig. 3b, c, $\eta_x = -1$ and $\eta_{z,w} = -1$. On the other hand, if the domain region (region (II) in Fig. 3a) consists of ($N_d + 1$) unit cells in the horizontal direction, the rotation angle of the polarization vector of the ($n_x$, $n_z$)-th unit cell of this region can be obtained as $\phi_z = \eta_{z,d} \pi(n_z/N_z + 1/2)$. Here, $\eta_{z,d}$ is the rotation direction of the helix in the $z$-axis direction of the domain region, and $\eta_{z,d} = +1$ as shown in Fig. 3d. Now, the remaining regions (III) and (IV) have the opposite direction of the chiral structure compared with the vortices of regions (I) and (II) described above, respectively (Fig. 3e). However, considering the constituting helices, it has the same $\eta_x, \eta_{z,w}$ and $\eta_{z,d}$ values because it is continuously connected throughout the supercell.

**Asymmetry ratios for various polar vortex arrays**. Using the phenomenological model described above, polar vortices with various morphologies can be considered. When the size of the supercell of the vortex array is 28 unit cells, Fig. 4a–c corresponds to the cases where the size of regions (I) and (III) is 3, 6, and 13 unit cells, respectively. The directions of the vortex and the chiral structure of the domain regions are the same. For these vortex

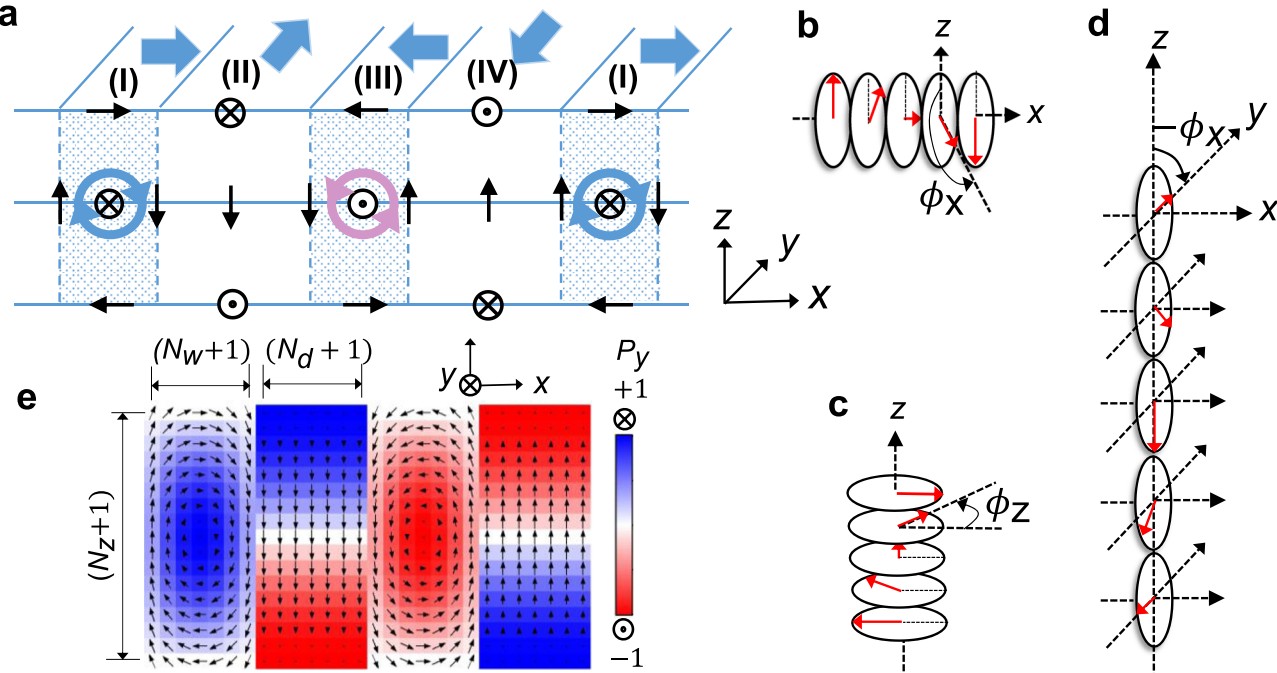

**Fig. 3 Phenomenological model of polar vortex array. a** Schematic of polar vortex array. Regions (I) and (III) consist of vortex pair connecting domain regions (II) and (IV). The bold blue arrows indicate the directions of polarizations of each region on the $(x–y)$ surface. The polarization vectors are uniform along the $y$-axis. **b**, **c** Helices about the $x$-axis (**b**) and $z$-axis (**c**) in the vortex-pair (I and III) regions. **d** A helix about the $z$-axis in the domain (II and IV) regions. **e** Structural parameters of polar vortex array in a phenomenological model.

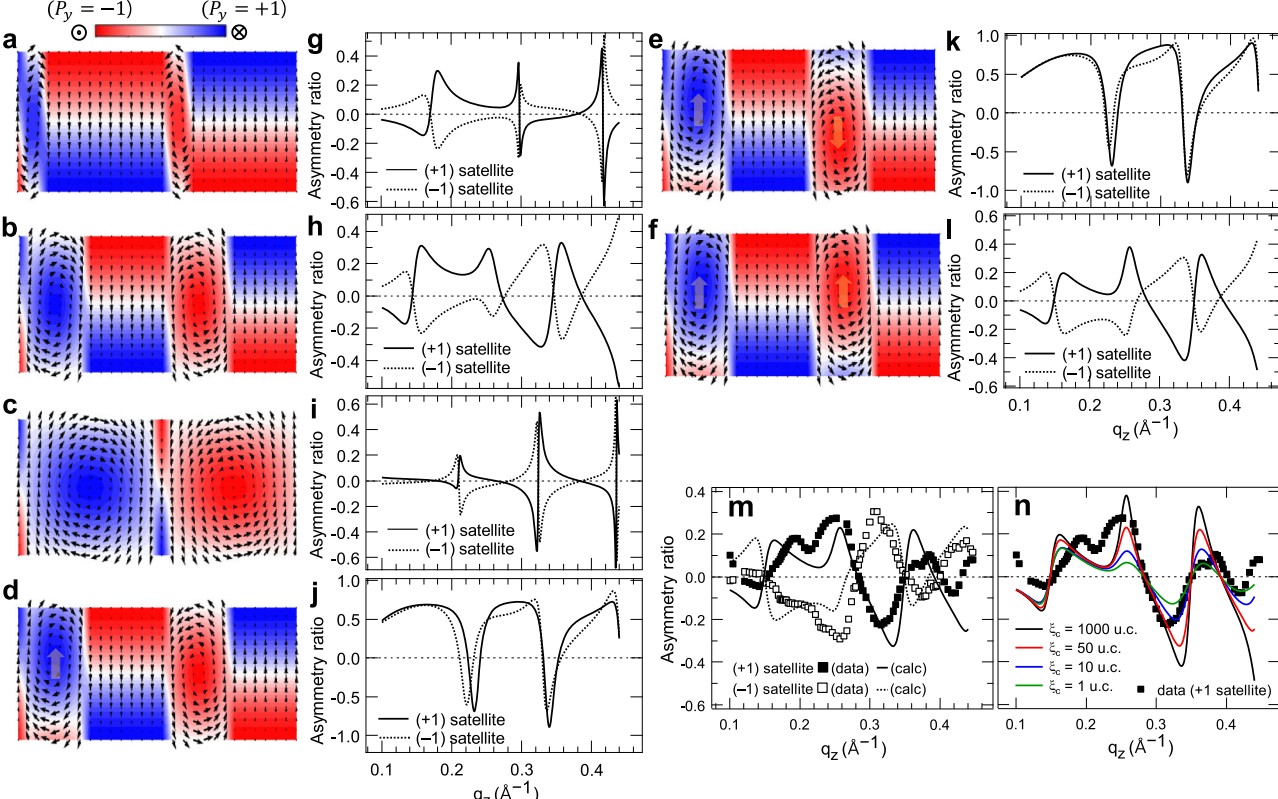

**Fig. 4 Asymmetry ratios of polar vortex arrays. a–f** Polar vortex pairs calculated using a phenomenological model in Fig. 3. For those with their core positions shifted from the center of the PbTiO3 layer, all shifts indicated by the arrows correspond to those by two unit cells. **g–l** Asymmetry ratios of $q_z$ rod scan intensities calculated at the lateral satellites with opposite signs for polar vortex pairs in **a–f**, respectively. **m** Asymmetry ratios of $q_z$ rod-scan intensities measured from a (PbTiO3)$n$/(SrTiO3)$n$ ($n = 16$ unit cells) superlattice on DyScO3 $(0\ 0\ 1)_{pc}$ substrate. The lines represent the calculation using the polar vortex pair model in **f** and **l** with the coherence length $\xi_c$ of 50 unit cells. **n** Effect of the coherence length $\xi_c$ explaining random fluctuation of polarization vectors.

arrays, we calculated the AR of $q_z$ rod scan using Eq. (12). The resonant energy is tuned to the $L_3$ $t_{2g}$ energy of the $Ti^{4+}$ ion where the maximum XCD appears as shown in Fig. 2g, and the AR of $q_z$ rod scan is calculated at the first satellite ($q_x = 0.057$ Å$^{-1}$) corresponding to the lateral period of the polar vortex array. The magnitude of the displacement vector $\mathbf{r}_n$ of the $Ti^{4+}$ atom was set to 10% of the lattice constant, which is the usual magnitude of the displacement vector of $Ti^{4+}$ atoms in perovskite ferroelectrics.

Figure 4g–i is the result of calculating the AR of REXS for the polar vortex array model of Fig. 4a–c, respectively. Two notable things can be seen here. First, when the size of the vortex region and the domain region is similar (Fig. 4h), the slope of the AR curve changes smoothly, whereas when one of two regions forms the majority (Fig. 4g, i), the sign of the AR curve changes abruptly at some $q_z$ values. As mentioned above, a helix in the $z$-axis direction that affects the $q_z$ dependence exists in each of the two regions. When the size of the two regions is similar, the AR curve is broader and smoother because scattering by two different helixes is mixed. On the other hand, if one region dominates, one helix is involved, so strongly coherent interference occurs in the depth direction, resulting in a very periodic derivative Gaussian shape.

Next, the ARs calculated at satellites with opposite signs are almost antisymmetric to each other (Fig. 4g–i), which is well known in the case of resonant magnetic scattering[12,36]. Because the sign of the satellite peak and the handedness of the helix have the same effect on the sign of the XCD, AR curves are antisymmetric for satellites of opposite sign. However, depending on the selected scattering geometry, the change in the diffraction angle to reverse the sign of the satellite peak cancels the sign change of the satellite peak by changing the polarization vector of the X-rays. Therefore, in this case, antisymmetric behavior in ARs may not appear[11].

In the resonant scattering of the polar vortex in this study, there is a separate condition in which ARs are clearly deviating from the antisymmetric behavior at the satellites of the opposite sign. This is when the core positions of the vortices are offset from each other (Fig. 4d, e). AR curves are not antisymmetric at all between satellites with opposite signs (Fig. 4j, k), but rather show almost the same values. In the case of vortices offset in the same direction (Fig. 4f), the antisymmetric relationship between opposite satellites is maintained. This demonstrates how REXS can sensitively probe small changes in the polarization texture. This sensitivity of AR curves is valid even if the difference in the position of the vortex core of the vortex pair is small within one unit cell. (see Supplementary Note 3).

On the other hand, when the vortex cores of the vortex pair move in opposite directions, and the difference in core position widens up to 4 unit cells (Fig. 4e), the AR curves for satellites with opposite signs become even more similar (Fig. 4k). These results suggest that the dominant factor in the antisymmetry between AR curves with opposite-sign satellites is the lateral helix penetrating the vortex pair (Fig. 3b) rather than the chiral structure of the vortex itself. That is, as the core positions of the vortex pair are shifted from each other, the coherence of the lateral helix is weakened, and the sign of the lateral satellite and the handedness of the lateral helix do not exactly match. From this point of view, it can be seen that Lovesey and van der Laan[18] modeled the polar vortex array as a one-dimensional helix in the lateral direction to well explain the antisymmetry of AR curves with respect to the opposite-sign satellite.

There are still important points to discuss about vortex pairs. According to the high-resolution STEM experimental results on the actual polar vortex array[5,6], the vortex pairs are more often observed that the core positions do not coincide with each other

in the depth ($z$) direction, whereas the results of REXS show that the AR curves of opposite-sign satellites are antisymmetric to each other[4,8]. To explain these apparently conflicting results, we have to consider another polar vortex array that corresponds to the mirror image of the original one. In a mirror image, the handedness of all helixes involved is reversed. Let us consider a polar vortex array in which the core position of the vortex pair is shifted by 1 unit cell and a mirror-reflected image along the lateral direction (Supplementary Fig. 3a). In the original image, all vortexes have a right-handed core, whereas in the mirror image, they have a left-handed core. The AR curve for the mirror image is exactly antisymmetric for the satellites with opposite signs among the AR curves of the original image (Supplementary Fig. 3b). According to this relationship, when vortex pairs with mirror images and handedness in opposite directions coexist in a sample, the AR curves obtained by incoherently summing the resonant-scattering intensities of the two cases are always antisymmetric for the opposite-sign satellite (Supplementary Fig. 3c). This result can well explain the experimental results in which the AR curves of opposite-signed satellites appear antisymmetric to each other, and this also explains the experimental results for the case of polar skyrmions.

**Comparison with experimental data.** We compare the calculated model with the actual experimental results. Figure 4m shows the AR of $q_z$ rod scan measured in the $(PbTiO_3)_n/(SrTiO_3)_n$ ($n = 16$ unit cells) superlattice. The X-ray energy was tuned to the $L_3$ $t_{2g}$ energy (around 457 eV) of the $Ti^{4+}$ atom where the XCD effect is maximized (Fig. 2g). Figure 4m shows that AR measured from the first-order satellites of opposite signs has an almost antisymmetric shape. This corresponds to the case where the cores of the vortex pair are located at the same depth, as discussed earlier.

Since the AR curve can be reproduced using a finite number of three-dimensional polar-texture models, it is difficult to completely explain the experimental data. As a result of comparing the AR calculations and experimental data for possible vortex-pair models according to these criteria, the case closest to the experimental data has vortex and domain regions with 6 and 8 unit cells, respectively, as shown in Fig. 4f, l, and the vortex core is shifted by 2 unit cells from the center of the PbTiO$_3$ layer toward the surface. Structural parameters used for best fits for REXS intensity in Fig. 2d and AR in Fig. 4m are $(N_w, N_x, N_z) = (7, 28, 15)$, $(\eta_x \cdot \eta_{zw}, \eta_{zd}) = (-1, -1, +1)$, and $(\xi_v, \xi_c) = (0.7\Lambda, 10$ unit cells), where the $(SrTiO_3/PbTiO_3)$ bilayer thickness $\Lambda = 114$ Å and the lattice parameters of the PbTiO$_3$ unit cell were $a_{PTO} = 3.832$ Å and $c_{PTO} = 3.8$ Å. The number of bilayers was 8, and the interfacial roughness was 4 Å. However, even when the core positions of the vortex pair are all at the center of the PbTiO$_3$ layer, the $q_z$ positions where the sign of the AR curve changes are well explained (Fig. 4b, h). Therefore, it may be relatively less important that the core positions of the vortex pair should be shifted to the same height. Considering this, it can be seen that the vortex-pair model that best explains the experimental data is very similar to the result obtained by the second-principles calculations[4,8].

To better explain the actual experimental data, we consider the random fluctuation of the polarization vector mentioned previously. The variables that determine the random fluctuation include the fluctuation amplitude $\sqrt{\langle \delta r^2 \rangle}$ and the coherence length $\xi_c$. In order for the amplitude $\sqrt{\langle \delta r^2 \rangle}$ to have a clear effect on the AR curve, it must be greater than the magnitude $|\mathbf{r}_n|$ of the displacement vector of $Ti^{4+}$ atom (Fig. 1b–e). This case is excluded because it violates the assumption of small fluctuation,

and the amplitude is fixed to $\sqrt{\langle \delta r^2 \rangle} = 0.1 |\mathbf{r}_n|$. On the other hand, the shorter the coherence length $\xi_c$, the smaller the amplitude of the AR at higher $q_z$ and the smoother the slope of the AR curve can be seen. The best fit in Fig. 4n is when the coherence length $\xi_c$ corresponds to 50 unit cells. Comparing with the periodicity of a polar vortex array that is 28 unit cells, it can be seen that at least one supercell formed of a vortex pair maintains a coherent shape.

## Discussion

We presented a detailed theoretical framework and practical calculation method of the REXS study to elucidate the polar chiral structure. First, the calculation of the REXS amplitude based on AT scattering for the resonant atoms with the polarization vector and the calculation of the structure factor was described in detail for the polar vortex array formed in $PbTiO_3/SrTiO_3$ superlattices. From these calculations, we obtained quantitatively the AR curve of the $q_z$ rod scan and compared it with the experimental data to show very similar results to the vortex-pair texture obtained by the second-principles calculations. In particular, the random fluctuation effect of the polarization vectors and the vertical correlation effect in the superlattice phase factor were introduced to better explain the actual data.

We introduced a phenomenological model based on the results of the second-principles calculations to determine the three-dimensional polar vortex pair texture with several structural parameters, and then calculated the AR of the $q_z$ rod scan for this vortex array model and compared it with the experimental data. However, if it is difficult to find a simple phenomenological model, a method to analyze resonant scattering in parallel with phase-field modeling is required. In particular, recent ultrafast time-resolved X-ray diffraction studies on the terahertz pulse response of polar vortex arrays have performed phase-field simulations by solving time-dependent equations of free energy to understand dynamics[9]. Although the study was limited to the nonresonant hard X-ray diffraction technique, it is expected that REXS will be able to clearly identify the three-dimensional vortex texture that changes in subpicoseconds by using the theoretical framework and analysis method developed in this study and combining phase-field modeling.

Another reason why quantitative analysis of REXS for the electric-polarization vector is important is to comprehensively analyze REXS with contributions from electric and magnetic channels simultaneously, such as the recently reported REXS study on electric and magnetic chiral structures of multiferroic $BiFeO_3$[17]. This study is expected to be an important characterization tool for identifying the mechanism by which multiferroics-based electronic or spintronic devices operate. In particular, X-ray resonant scattering is essentially the only method that can simultaneously measure the response of both electric polarization and magnetic moment to an external signal.

## Methods

**Materials' growth.** Superlattices of $(PbTiO_3)_n/(SrTiO_3)_n$ ($n = 16$ unit cells) were deposited on $DyScO_3$ (110)-oriented substrates using pulsed-laser deposition. The growth temperature was 620 °C, and the oxygen pressure was 100 mTorr. Growth was monitored with reflection high-energy electron diffraction (RHEED) to monitor the layer-by-layer, epitaxial growth. Following deposition, the superlattice was annealed in 50 Torr of oxygen for 10 minutes and then cooled to room temperature.

**Synchrotron X-ray diffraction.** Synchrotron X-ray diffraction was performed at the Sector 33-BM-C beamline at the Advanced Photon Source, Argonne National Laboratory, USA. Using a four-circle Huber diffractometer and a PILATUS 100 K pixel-area detector, 3D reciprocal space maps were obtained in order to observe peaks that arise from the superlattice and polar vortex ordering.

**Resonant elastic X-ray scattering.** REXS experiments were performed at beamline 4.0.2 at the Advanced Light Source, beamline 29-ID-C at the Advanced Photon Source, and beamline I0 at the Diamond Light Source. The X-ray energy was tuned to the Ti $L_{2,3}$ edge, and the diffracted intensity was measured using an in-vacuum CCD area detector sensitive to soft X-rays. The diffracted intensity was measured for both right- and left-circularly polarized X-rays. Dark images (images with no X-ray exposure) were taken and used to subtract background counts of the detector. To obtain data, the diffraction-peak intensity measured on the detector was used, and a background region on the detector was used to subtract the fluorescence background from the diffraction peak. Circular dichroism was measured by taking the difference between measurements with right- and left-circularly polarized X-rays. For constant-$q$ energy scans, the energy was scanned around 457–470 eV to observe the $L_{2,3}$ edge, and the sample and detector angles were adjusted to maintain scattering at a constant momentum transfer. For constant-energy reciprocal-space scans, the energy was chosen where the maximum circular dichroism occurs (around 457.2 eV), and the sample and detector angles were adjusted to maintain a constant lateral scattering vector while changing the out-of-plane scattering vector.

**First-principles calculations.** We performed first-principles density-functional-theory calculations with local density approximation[37,38]. The atomic structures of $PbTiO_3$ were calculated using the Vienna ab initio simulation package (VASP)[39,40] with the projector-augmented wave (PAW) method[41]. The energy cut-off of 500 eV and the $k$-point sampling on an $8 \times 8 \times 8$ grid were used with a 5-atom unit cell. The anisotropic tensors were calculated using the WIEN2k package[42,43] using the full-potential augmented plane-wave plus local orbital methods. The $k$-point sampling on a $12 \times 12 \times 12$ grid was used. We calculated the core-level energies, band structures for conduction bands, and the matrix elements between the Ti $L_{2,3}$ core levels and conduction bands[27], which are used to evaluate the anisotropic tensor using Eq. (3) with $\Gamma$ of 0.2 eV.

**Reporting summary.** Further information on research design is available in the Nature Research Reporting Summary linked to this article.

## Data availability

All the data supporting this work are available from the corresponding author upon reasonable request.

## Code availability

The code used for the simulations in this work is available from the corresponding author upon reasonable request.

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

## Acknowledgements

K.T.K., S.Y.P., and D.R.L acknowledge financial support by National Research Foundation of Korea (Grant No. NRF-2020R1A2C1009597, NRF-2019K1A3A7A09033387, and NRF-2021R1C1C1009494). M.M. and R.R. were supported by the Quantum Materials program from the Office of Basic Energy Sciences, US Department of Energy (DE-AC02-05CH11231). V.A.S., J.W.F., and L.W.M. acknowledge the U.S. Department of Energy, Office of Science, Office of Basic Energy Sciences, under Award Number DE-SC-0012375 for support to study complex-oxide heterostructure with X-ray scattering. L.W.M. and R.R. acknowledge partial support from the Army Research Office under the ETHOS MURI via cooperative agreement W911NF-21-2-0162. J.Í. acknowledges financial support from the Luxembourg National Research Fund through project FNR/C18/MS/12705883/REFOX. M.A.P.G. was supported by the Czech Science Foundation (project no. 19-28594X). Diamond Light Source, UK, is acknowledged for beamtime on beamline I10 under proposal NT24797. Use of the Advanced Light Source, Lawrence Berkeley National Laboratory, was supported by the U.S. Department of Energy (DOE) under contract no. DE-AC02-05CH11231, and use of the Advanced Photon Source was supported by DOE's Office of Science under contract DE-AC02-06CH11357.

## Author contributions

M.R.M., V.A.S., S.D., C.K., E.P.D., D.M.B., P.S., F.R., S.W.L., G.v.d.L., and J.W.F. performed the REXS experiments. M.R.M., V.A.S., and J.W.F. carried out the nonresonant, hard X-ray diffraction experiments. M.A.P.G., F.G.-O., J.Í., P.G.-F., and J.J. performed the second-principles calculations. S.D. grew the superlattices using pulsed-laser deposition. S.S. took the STEM image. K.T.K. and D.R.L. performed the REXS calculations. S.Y.P. performed the first-principles calculation. J.W.F., R.R., and D.R.L. supervised the study. K.T.K., M.R.M., V.A.S., S.Y.P., G.v.d.L., L.W.M., J.W.F., R.R., and D.R.L. cowrote the paper. All authors contributed to the discussion and paper preparation.

## Competing interests

The authors declare no competing interests.
