## [Peer review file · Nature Communications]

REVIEWER COMMENTS

Reviewer #1 (Remarks to the Author):

The authors describe a soft x-ray resonant scattering simulation of a PTO/STO system with a complex polarization configuration and a comparison of the results with an experimental study. The calculation consists of a parameterization of the resonant scattering using a model system incorporating a similar ionic environment and a detailed model of the polarization distribution. The field of complex polarization configurations is of significant interest and poses a significant characterization challenge that the authors address. Further development could describe states such as Skyrmions that are of even broader interest. The manuscript, however, has several significant issues that the authors should address. My overall opinion is that the work would be suitable for publication if the authors can address them.

1. Overall, the predictions, and recaps of previous TEM and hard x-ray scattering measurements occupy seven of the eight figures. The comparison of the simulation and data is cursory and qualitative. First, the parameters for the simulation shown in Fig. 8 are not summarized clearly. The parameters used to find the best fit in Fig. 8(a) should be clearly given. Figure 8a, in particular, does not have the parameters clearly defined and the reader has to guess what values have been used. Does Fig. 8, for example, consider the case with the mirrored polarizations? Does Fig. 8 include the finite vertical correlation length described in the supplement?
2. Along the same lines as point 1, the relatively poor agreement with the q_z dependence in Fig. 8 suggests that the structural model may not accurately describe the z dependence of the polarization. Does a comparison of the simulated non-resonant intensity with the hard x-ray reciprocal space map yield similar disagreement? Along the same lines, the soft x-ray reflectivity of the +1 satellite shows an apparently short correlation length but is highly complicated because of the short penetration depth at soft x-ray wavelengths. The photon energy for supplemental figure 2 is also not given. Does the hard x-ray reciprocal space map show a similarly short correlation length?
3. It is not clear why rotating the PZT scattering tensor is the right way to describe Ti displacements that are not along the c axis. The authors should explain in more detail why this approach is valid.
4. The TEM image does not appear to show a polarization distribution that matches the diagram in panel (b) of the second Fig. 1 or the simulated distributions in Fig. 5. The authors should comment on the origin of this difference, for example structural differences in the samples or elastic differences between TEM and x-ray studies.

5. Finally, and most importantly, the manuscript does not clearly describe which aspects of the simulation work are novel. The readership of Nature Communications is very broad and would benefit from a clear emphasis on the most innovative part of the work in the introduction. The correlation length and similar issues are important in interpreting the results, but not of real general interest. I found the most useful and innovative part of the work to be the adoption of appropriate components of T0 and the simulation of a meaningful AR but found this novelty hard to find in the introduction.

There are also several details that I would like to ask the authors to address:

6. There are two figure captions labeled Fig. 1.

7. I found the color schemes in the figures to be very distracting. Is there a significance to the choice of a red-blue gradient in Fig. 3 and a purple-green gradient in the second Fig. 1?

8. I found the perspective view in Fig. 3(d) very hard to understand. It's not clear what the variation along the y axis is supposed to be.

9. The supplement says: "Intensities were measured at the lateral satellite peak ($q_x=0.057 \text{ \AA}^{-1}$) corresponding to the polar vortex array periodicity and at the 2nd order multilayer peak ($q_z=0.164 \text{ \AA}^{-1}$)." Does this mean they were measured twice or is this one set of coordinates, i.e. a since (q_x, q_z) pair?

10. There are several sentences for which the meaning is not clear. For example, the meaning of the sentence in lines 348 to 351 is not clear. Phrases such as "is relevant to describe" and "well consistent" are awkward and would benefit from clarification.

11. It would be useful to the reader to have the components of T0 given in the supplement.

Reviewer #2 (Remarks to the Author):

The manuscript “Chiral structures of electric polarization vectors quantified by x-ray resonant scattering” by Kook Tae Kim et al. reports an elaborate computational analysis of an exotic electric polarization domain vortices in a ferroelectric/paraelectric superlattice thin film. This computational analysis is used in the manuscript to describe soft x-ray resonant scattering results.

Within the proposed framework, the results appear to be valid, and the discussion of various polar vortex configurations is thoughtful and interesting. The findings are well presented and clearly described.

The significance of the results, however, is limited to a very narrow class of ferroelectric multilayer systems studied with a very special kind of tools, namely, soft x-ray resonant scattering with circularly polarized x-rays. It is important to discuss significance of the computational approach beyond the proof of concept.

Despite using resonant scattering and soft x-rays, the authors do not include the energy-dependent atomic scattering factors and absorption effects. Is this approximation justified considering the attenuation depth of soft x-rays may be comparable to the size of polarization domains?

Do the authors have x-ray scans for both +1 (given in supplementary materials) and -1 in-plane satellite scattering peak? I suggest the authors include soft x-ray diffraction scan(s) shown within the main body of the manuscript.

Unfortunately, I cannot recommend the manuscript for publication until the concerns of significant and attenuation depth at low (and resonant) x-ray energies are addressed.

We greatly appreciate the reviewers' valuable comments on our manuscript. We have revised the manuscript to address the comments made by the reviewers. We mostly agree with the reviewers' comments and have implemented changes accordingly. The comments from the reviewers (in blue) and our response (in black) and changes/actions (in red) are given below.

Reviewer #1

1. Overall, the predictions, and recaps of previous TEM and hard x-ray scattering measurements occupy seven of the eight figures.

→ Since the main purpose of this study is to present a method of quantitative analysis of x-ray resonant scattering, which has many advantages over TEM and non-resonant hard x-ray scattering, most of the figures were compared with the existing experimental results. . However, as the reviewer pointed out and suggested below, the revised manuscript was reconstructed focusing on the most important and innovative topic of how to calculate the resonant scattering amplitude or anisotropic tensor T_{ij} , which is the starting point for quantitative analysis. Accordingly, new figures for T_{ij} calculations were added, energy dependence in supplementary information was moved to the figure in the main text, and relatively less important simulation results were moved to supplementary information. Therefore, the figures in the body of the revised manuscript contain more diverse contents.

The comparison of the simulation and data is cursory and qualitative. First, the parameters for the simulation shown in Fig. 8 are not summarized clearly. The parameters used to find the best fit in Fig. 8(a) should be clearly given. Figure 8a, in particular, does not have the parameters clearly defined and the reader has to guess what values have been used. Does Fig. 8, for example, consider the case with the mirrored polarizations? Does Fig. 8 include the finite vertical correlation length described in the supplement?

→ The parameter values used for the best fit in Fig. 8(a), including the finite vertical correlation length, were added to the text as follows.

"Structural parameters used for best fits for REXS intensity in (revised) Fig. 2d and AR in (revised) Fig. 4m are $(N_w, N_x, N_z) = (7, 28, 15)$, $(\eta_x, \eta_{zw}, \eta_{zd}) = (-1, -1, +1)$, and $(\xi_v, \xi_c) = (0.7 \Lambda, 10 \text{ unit cells})$, where the $(\text{SrTiO}_3/\text{PbTiO}_3)$ bilayer thickness $\Lambda = 114 \text{ \AA}$ and the lattice parameters of the PbTiO_3 unit cell were $a_{\text{PTO}} = 3.832 \text{ \AA}$ and $c_{\text{PTO}} = 3.8 \text{ \AA}$. The number of bilayers was 8, and the interfacial roughness was 4 \AA ."

Figure 8 is a case where mirrored polarization is not considered. The sample in the case of

incoherently coexisting mirrored polarizations with opposite handedness mentioned in the text below is different from the sample in Figure 8. The two samples have different thicknesses of the PTO layer. This point is also discussed in reviewer's comment #4 below. In order to avoid unnecessary confusion, the contents of the main text have been partially removed as follows.

"This corresponds to the case where the cores of the vortex pair are located at the same depth, or in the case of incoherent sums with different core positions. The latter refers to a case in which vortex pairs with opposite handedness of the vortex core exist incoherently. This is a reasonable explanation for the experimental data, and recent four-dimensional STEM measurements have demonstrated the coexistence of vortex pairs with opposite handedness.³⁴" → "This corresponds to the case where the cores of the vortex pair are located at the same depth, as discussed earlier."

2. Along the same lines as point 1, the relatively poor agreement with the q_z dependence in Fig. 8 suggests that the structural model may not accurately describe the z dependence of the polarization. Does a comparison of the simulated non-resonant intensity with the hard x-ray reciprocal space map yield similar disagreement? Does the hard x-ray reciprocal space map show a similarly short correlation length?

→ As mentioned in the text, the advantages of the AR is that the interface roughness effect and the superlattice structure factor, which are not related to the polar vortex, are canceled out in the denominator and the numerator. Therefore, the fitting of the AR in Fig. 8 is not improved by a more accurate determination of superlattice structure factor including the vertical correlation of polar vortex arrays between different PTO layers. To improve the fitting of the AR curves, more accurate information about the chiral structure of polar vortex array is needed. Since non-resonant hard x-ray scattering is insensitive to the chiral structure of electric polarization vectors, it is not expected to improve the fitting of the q_z dependence of the AR in Fig. 8.

Instead of performing simulations on non-resonant scattering intensity, we intend to answer the reviewer's question by discussing the line profiles of the hard x-ray reciprocal space map. For a clearer discussion, the RSM around the DyScO₃ (002)_{pc} peak shown in Fig. 2 (second Fig. 1) was replaced with the RSM around the (004)_{pc} peak. (revised Fig. 2a) We compare the vertical line profiles from the RSM corresponding to the specular scan ($q_x = 0$) crossing the DyScO₃ (004)_{pc} peak and the q_z rod scans crossing the satellite peaks ($q_x = \pm 0.057 \text{ \AA}^{-1}$), respectively. (Revised Fig. 2b) In the case of the specular scan, the total thickness fringes of the superlattice are clearly seen between the superlattice peaks, whereas only the broad superlattice peaks were observed in the case of the q_z rod scan. If the vortex array is formed coherently over the entire superlattice from the surface to the substrate, the q_z rod scan should show the total thickness fringes like the

specular scan because the vertical correlation length is longer than the total thickness of the superlattice. However, since the thickness fringe is not seen in the line profile of the q_z rod scan, the vertical correlation length can be considered to be shorter than the total thickness. The content discussed above is included in the text as follows, and figures have been added.

(Revised Fig. 2b)

“Comparing the vertical line profiles crossing the DyScO₃ (004)pc peak and the lateral satellite peaks, there is a clear difference. (Fig. 2b) In the former, the total thickness fringe of the superlattice is clearly visible, whereas in the latter, only broad superlattice peaks are observed. This indicates that the vertical correlation length does not reach the total thickness because the polar vortex arrays of the PbTiO₃ layers separated by the SrTiO₃ layers are structurally different from each other or do not align coherently in the vertical direction.”

Along the same lines, the soft x-ray reflectivity of the +1 satellite shows an apparently short correlation length but is highly complicated because of the short penetration depth at soft x-ray wavelengths. The photon energy for supplemental figure 2 is also not given.

→ The q_z rod scan of soft x-ray resonant scattering in supplemental figure 2 was measured at 456.8 eV (refer to supplemental figure 1), which is the resonant energy at which XCD is maximized. Therefore, as the reviewer points out, it can be complicated to determine the vertical correlation length because of the effect of the short penetration length of soft x-rays.

Although we did not explicitly include the absorption effect, since the anisotropic tensor T_{ij} itself already includes the absorption effect, it is only necessary to explicitly include the absorption effect in the superlattice structure factor of Equation (8). In particular, since the electric polarization vector does not appear in the SrTiO₃ layer, it does not contribute to the AT, but the absorption effect due to the Ti atom must be considered. In order to consider the absorption effect in Equation (8), it is sufficient to describe q_z by replacing it with the complex variable \tilde{q}_z . For this purpose, the contents of the text have been modified as follows.

$$F_{sup,\epsilon'\epsilon} = F_{0,\epsilon'\epsilon} + F_{1,\epsilon'\epsilon}e^{i\tilde{q}_z\Lambda} + F_{2,\epsilon'\epsilon}e^{i\tilde{q}_z2\Lambda} + \dots + F_{N-1,\epsilon'\epsilon}e^{i\tilde{q}_z(N-1)\Lambda} \quad (8)$$

“To include the absorption effect, a complex variable $\tilde{q}_z = (4\pi/\lambda)\sqrt{n_{eff}^2 - \cos^2\theta_{in}}$ (θ_{in} is the incident angle and λ is the x-ray wavelength) is used. The imaginary part of the x-ray refractive index n_{eff} corresponds to the absorption effect. In the case of SrTiO₃/PbTiO₃ superlattice, effective index $n_{eff} = (n_{STO}t_{STO} + n_{PTO}t_{PTO})/(t_{STO} + t_{PTO})$, where $n_{STO(PTO)}$ and $t_{STO(PTO)}$ are the refractive index and the layer thickness of the SrTiO₃(PbTiO₃) layer, respectively.”

In this case, Equation (9) is changed as follows.

$$|F_{sup,\epsilon'\epsilon}|^2 = |F_{\epsilon'\epsilon}|^2 \sum_{k=0}^{N-1} \left(e^{-\text{Im}[\tilde{q}_z]\Lambda k} + \sum_{k'=k+1}^{N-1} 2 \cos(\text{Re}[\tilde{q}_z]\Lambda(k' - k)) e^{-\frac{|k-k'|}{\xi_v}} e^{-\text{Im}[\tilde{q}_z]\Lambda(k+k')} \right) \quad (9)$$

In the case of the soft x-ray q_z rod scan, quantitative calculations can be obtained using equations (8)-(9) modified to consider the absorption effect. As shown in the figure (new Fig. 2d) below, considering the absorption effect, the total thickness fringe is not seen well, so the penetration depth at the resonant energy (456.8 eV) is only a part of the superlattice. However, in order to explain the amplitude of oscillation of the q_z rod scan of the experimental data, it was confirmed that the vertical correlation length did not exceed one bilayer.

(Revised Fig. 2d)

“On the other hand, the q_z rod scan intensity calculated using equations (8)-(9) considering the absorption effect shows that the effective penetration depth is shorter than the total thickness because the total thickness fringe is not clearly visible even when the vertical correlation length ξ_v is the same as the total thickness of the superlattice.(Fig. 2d) However, in order to explain the amplitude of oscillation of the experimental data, it can be seen that the vertical correlation length should not exceed one bilayer.”

3. It is not clear why rotating the PZT scattering tensor is the right way to describe Ti displacements that are not along the c axis. The authors should explain in more detail why this approach is valid.

→ We found that the rotation method is a good approximation after comparing the results of first-principles calculation for the case of Ti displacement in an arbitrary direction with the case of rotating the anisotropy tensor, which is the resonant scattering amplitude. **More detailed explanations and figures (revised Fig. 1b and c-i) have been added to the text as a new subsection with a title of "Anisotropic tensor of the resonant ion with polarization pointing in an arbitrary direction", and information about first-principles calculation has been also added to the Method section.**

4. The TEM image does not appear to show a polarization distribution that matches the diagram in panel (b) of the second Fig. 1 or the simulated distributions in Fig. 5. The authors should comment on the origin of this difference, for example structural differences in the samples or elastic differences between TEM and x-ray studies.

→ As the reviewer pointed out, the TEM images are from reference [1] and were obtained from the superlattice of [STO_n/PTO_n] (n = 10 unit cells), whereas the sample used in this study was the superlattice of [STO_n/PTO_n] (n = 16 unit cells), so there is a distinct structural difference. **This information has been explicitly added to the text and figure captions.**

Despite these structural differences, the reciprocal space map of non-resonant x-ray scattering and XCD of soft x-ray resonant scattering for both samples showed similar results, so it was presented as a typical example of a polar vortex array. In addition, considering the limitations of the phenomenological model presented in this text, it can be seen that the structural features of the simulation distribution in revised Fig. 4(b) are similar to those of the TEM image.

5. Finally, and most importantly, the manuscript does not clearly describe which aspects of the simulation work are novel. The readership of Nature Communications is very broad and would benefit from a clear emphasis on the most innovative part of the work in the introduction. The correlation length and similar issues are important in interpreting the results, but not of real general interest. I found the most useful and innovative part of the work to be the adoption of appropriate components of T₀ and the simulation of a meaningful AR but found this novelty hard to find in the introduction.

→ The core topic of this study is to provide a theoretical framework to quantitatively analyze the x-ray resonant scattering intensity of the chiral structures of electric polarization vectors. However, as the reviewer pointed out, the most useful and innovative parts are (1) for the first time, we propose a method for quantitatively obtaining the component of the resonant scattering amplitude tensor T_{ij} for a resonant atom with an electric polarization vector directed in an

arbitrary direction, and (2) systematically show how sensitive REXS is to the vortex pair structure through quantitative simulation of the asymmetry ratio for the polar vortex. In particular, in order to solve the former, the resonant scattering amplitude tensor T_n for a polarization vector directed in an arbitrary direction must be obtained quantitatively. However, as in the answer to reviewer's comment #3 above, it is absolutely necessary to verify whether the approximation obtained by rotating T_0 of the basis atom presented here is valid. To emphasize this point, **the introduction, abstract, and conclusion have been revised as follows.** In particular, compared to the previous studies of Lovesey and van der Laan (reference [17]), our study clearly clarifies that there is a breakthrough and shows why the results of this study are beneficial to broad readers.

In Abstract, "To this end, a quantitative calculation method of the anisotropic tensor, which is the REXS amplitude for an electric polarization vector pointing an arbitrary direction, is presented in detail." "Based on the theoretical framework established here, REXS for polar chiral structures will become a useful and popular tool as x-ray resonant magnetic scattering (XRMS) is intensively used in the study of magnetic chiral domains and skyrmions. Furthermore, it will enable a quantitative and comprehensive study of both electric and magnetic REXS on the chiral structures of multiferroic materials."

In Introduction: "In addition, for technological application of polar chiral structures, non-destructive *in situ* characterization of structural responses to electrical, optical, and mechanical external signals is required."

"However, they obtained the AT value for the polarization vector in an arbitrary direction by rotating the AT of the basis atom, but there was no verification of whether this assumption was valid. This assumption is valid in TTS because, in the case of a single crystal, the site symmetry of the resonant atoms are all the same and their surroundings can always be expressed as rotational operations. On the other hand, in the case of a polar chiral structure formed in a thin film, since the surrounding environment of resonant atoms at different positions changes gradually, site symmetry is very low, making it difficult to describe it as a rotational operation. In addition, it is difficult to quantitatively analyze experimental data because the method for determining the AT component value of the basis atom is not yet known."

"Here, we present a detailed theoretical framework to quantitatively analyze the x-ray resonant scattering intensity from the chiral structure of electric polarization vectors. First, we present a method to quantitatively obtain the component of the resonant scattering amplitude AT for a resonant atom with an electric polarization vector directed in an arbitrary direction. In particular, we verify whether the approximation obtained by rotating the AT of the basis atom is valid by

comparing it with the first-principles calculation, and also present a method to quantitatively obtain the basis AT."

In Discussion: "Another reason why quantitative analysis of REXS for electric polarization vector is important is to comprehensively analyze REXS showing electric and magnetic channels simultaneously, such as the recently reported REXS study on electric and magnetic chiral structures of multiferroic BiFeO₃. This study is expected to be an important characterization tool for identifying the mechanism by which multiferroics-based electronic or spintronic devices operate. In particular, x-ray resonant scattering is essentially the only method that can simultaneously measure the response of both electric polarization and magnetic moment to an external signal."

6. There are two figure captions labeled Fig. 1.

→ We added new figures and regrouped the figures to clarify the connection between them, and we have corrected the figure captions accordingly.

7. I found the color schemes in the figures to be very distracting. Is there a significance to the choice of a red-blue gradient in Fig. 3 and a purple-green gradient in the second Fig. 1?

→ There is no significance in the choice of a color gradient. We use the same color gradient in the revised manuscript.

8. I found the perspective view in Fig. 3(d) very hard to understand. It's not clear what the variation along the y axis is supposed to be.

→ We redrew the schematic in Figure 3(d) to clarify the rotation in the y-z plane. (revised Fig. **3d**)

9. The supplement says: "Intensities were measured at the lateral satellite peak ($q_x=0.057 \text{ \AA}^{-1}$) corresponding to the polar vortex array periodicity and at the 2nd order multilayer peak ($q_z=0.164 \text{ \AA}^{-1}$)." Does this mean they were measured twice or is this one set of coordinates, i.e. a since (q_x, q_z) pair?

→ This is one set of coordinates, a single (q_x, q_z) pair. In order to avoid unnecessary misunderstanding, the sentence has been modified as follows.

In revised Fig. 2f-g, "Intensities were measured at the lateral satellite peak position ($q_x = 0.057 \text{ \AA}^{-1}$, $q_z = 0.164 \text{ \AA}^{-1}$)"

10. There are several sentences for which the meaning is not clear. For example, the meaning of the sentence in lines 348 to 351 is not clear. Phrases such as "is relevant to describe" and "well consistent" are awkward and would benefit from clarification.

→ We identified the sentence pointed out by the reviewer as being in the abstract. As previously pointed out by the reviewer in comment #5, these sentences were removed as the content of the abstract changed.

11. It would be useful to the reader to have the components of T_0 given in the supplement.

→ We added a figure showing the energy dependence of the components T_{xx} and T_{zz} of T_0 . (revised Fig. 2e) As in the answer to reviewer's comment #5, the calculation of T_n is important, and at the same time, it is also important to quantitatively determine the component of T_0 . We fully agree with the reviewer's suggestion, and the content in the Supplementary Information with the title of "Basis of AT determination measurement" has been moved to the main text with figures (revised Fig. 2e-g).

Reviewer #2

The significance of the results, however, is limited to a very narrow class of ferroelectric multilayer systems studied with a very special kind of tools, namely, soft x-ray resonant scattering with circularly polarized x-rays. It is important to discuss significance of the computational approach beyond the proof of concept.

→ We note that the emergence of such polar textures in ferroelectrics has essentially led to a “parallel universe” to magnetic spin textures. Thus, understanding the microscopic details of the formation of such polar vortices and skyrmions is of strong fundamental interest. Particularly, probing the details of the emergent chirality in such polar textures will give us. We want to emphasize that there have been numerous XRMS-related papers published over the past 30 years since the late 1980s, when the quantitative description of x-ray resonant magnetic scattering began. Likewise, it is expected that the quantitative analysis of x-ray resonant scattering for electric polarization vectors presented for the first time in this paper will be used intensively in structural analysis of emerging nanostructures of ferroelectric as well as multiferroic materials in the near future.

Please also see the answer to comment #5 of the previous reviewer (#1).

Despite using resonant scattering and soft x-rays, the authors do not include the energy-dependent atomic scattering factors and absorption effects. Is this approximation justified considering the attenuation depth of soft x-rays may be comparable to the size of polarization domains?

→ As the reviewer pointed out, we did not explicitly include the absorption effect. However, since the atomic scattering factor for the resonant atom corresponds to the anisotropic tensor T_{ij} , which is the resonant scattering amplitude, the energy-dependent atomic scattering factor is already properly included. To make this more clear, the energy dependence of T_{xx} and T_{zz} obtained from XAS and XLD of PZT, which was used to obtain the results of supplemental figure 1, was added to revised Fig. 2e.

Since the anisotropic tensor T_{ij} itself already includes the absorption effect, it is only necessary to explicitly include the absorption effect in the superlattice structure factor of Equation (8). In particular, since the electric polarization vector does not appear in the SrTiO₃ layer, it does not

contribute to the AT, but the absorption effect due to the Ti atom must be considered. In order to consider the absorption effect in Equation (8), it is sufficient to describe q_z by replacing it with the complex variable \tilde{q}_z . For this purpose, the contents of the text have been modified as follows.

$$F_{sup,\epsilon'\epsilon} = F_{0,\epsilon'\epsilon} + F_{1,\epsilon'\epsilon} e^{i\tilde{q}_z\Lambda} + F_{2,\epsilon'\epsilon} e^{i\tilde{q}_z2\Lambda} + \dots + F_{N-1,\epsilon'\epsilon} e^{i\tilde{q}_z(N-1)\Lambda} \quad (8)$$

“To include the absorption effect, a complex variable $\tilde{q}_z = (4\pi/\lambda) \sqrt{n_{eff}^2 - \cos^2 \theta_{in}}$ (θ_{in} is the incident angle and λ is the x-ray wavelength) is used. The imaginary part of the x-ray refractive index n_{eff} corresponds to the absorption effect. In the case of SrTiO₃/PbTiO₃ superlattice, effective index $n_{eff} = (n_{STO}t_{STO} + n_{PTO}t_{PTO})/(t_{STO} + t_{PTO})$, where $n_{STO(PTO)}$ and $t_{STO(PTO)}$ are the refractive index and the layer thickness of the SrTiO₃(PbTiO₃) layer, respectively.”

In this case, Equation (9) is changed as follows.

$$|F_{sup,\epsilon'\epsilon}|^2 = |F_{\epsilon'\epsilon}|^2 \sum_{k=0}^{N-1} \left(e^{-\text{Im}[\tilde{q}_z]\Lambda k} + \sum_{k'=k+1}^{N-1} 2 \cos(\text{Re}[\tilde{q}_z]\Lambda(k' - k)) e^{-\frac{|k-k'|}{\xi_v}} e^{-\text{Im}[\tilde{q}_z]\Lambda(k+k')} \right) \quad (9)$$

Although this result considers the absorption effect, Since the term in the parenthesis in equation (9) corresponds to the actual superlattice structure factor and is independent of the X-ray polarization vectors ϵ' and ϵ , it is sufficient to consider only the polar vortex supercell structure factor $F_{\epsilon'\epsilon}$ of one layer of PbTiO₃ for the asymmetry ratio.

Do the authors have x-ray scans for both +1 (given in supplementary materials) and -1 in-plane satellite scattering peak? I suggest the authors include soft x-ray diffraction scan(s) shown within the main body of the manuscript.

→ As suggested by the reviewer, supplemental figure 2 with soft x-ray q_z rod scan intensity was moved to the main body of the manuscript, and the q_z rod scan measured at the (-1)th order satellite peak was also added.(revised Fig. 2d)

In the case of soft x-ray q_z rod scan, quantitative calculations can be obtained using equations (8)-(9) modified to consider the absorption effect. As shown in the figure (revised Fig. 2d) below, considering the absorption effect, the total thickness fringe is not seen well, so the penetration depth at the resonant energy (456.8 eV) is only a part of the superlattice. However, in order to explain the amplitude of oscillation of the q_z rod scan of the experimental data, it was confirmed that the vertical correlation length did not exceed one bilayer.

(Revised Fig. 2d)

“On the other hand, the q_z rod scan intensity calculated using equations (8)-(9) considering the absorption effect shows that the effective penetration depth is shorter than the total thickness because the total thickness fringe is not clearly visible even when the vertical correlation length ξ_v is the same as the total thickness of the superlattice.(Fig. 2d) However, in order to explain the amplitude of oscillation of the experimental data, it can be seen that the vertical correlation length should not exceed one bilayer.”

Unfortunately, I cannot recommend the manuscript for publication until the concerns of significant and attenuation depth at low (and resonant) x-ray energies are addressed.

→ We emphasize that this study is not a simple extension of the well-known study of x-ray resonant scattering for magnetic moments, but presents for the first time a quantitative analysis of x-ray resonant scattering with a completely different mechanism for electric polarization vectors. In addition, we emphasize that this theoretical framework is a very important study, both academically and technologically, in that it opens new structural studies on the topological nanostructure of ferroic as well as multiferroic materials.

As for the attenuation effect, in addition to the previously presented kinematical calculations, dynamical calculation for electric field intensity distribution has been also added to the supplementary information. (Supplementary note 2 and supplementary Fig. 1)

In the text: “In order to maximize the XCD effect, the X-ray energy was tuned to the L_3 t_{2g} energy (around 457 eV) of the Ti^{4+} ion, which yields strong absorption effect. In this case, the penetration depth at which the x-ray intensity decreases by $1/e$ in the x-ray absorption spectroscopy (XAS) measurement may be only one $PbTiO_3$ layer close to the surface. However, in the case of soft x-ray scattering, the effective penetration depth can be much deeper because of the interference effect between the x-rays reflected from the interfaces. From the electric-field intensity

distributions as a function of incident angle and depth obtained for non-resonant (450 eV) and resonant (456.8 eV) energies using dynamical calculation, it can be seen that total thickness fringes are also seen in specular reflectivity even at the resonant energy when q_z is larger than 0.2 \AA^{-1} .(Supplementary Note 2 and Supplementary Fig. 1) For dynamical calculation of REXS intensities in non-specular q_z rod scan, the distorted-wave Born approximation (DWBA) developed for X-ray resonant magnetic scattering or grazing incidence small-angle X-ray scattering can be applied.”

In Supplementary Note 2 and Supplementary Fig. 1,

“As shown in the left panel in Supplementary Fig. 1(revised*), a distinct total thickness fringe can be seen in the specular reflectivity at non-resonant energy (450 eV) away from the absorption edge. Also, in the q_z region where the experimental data was measured, it can be seen that the electric field intensities in the PTO/STO superlattice are 10% or more of the incident beam. On the other hand, in the 456.8 eV (right panel) where the resonant scattering experiment (revised* Fig. 4m and revised* Fig. 2d) was performed, the total thickness fringe begins to be seen only in the q_z region greater than 0.2 \AA^{-1} . Convolution by angular divergence of 0.05 degree and interfacial roughness of 4 Å were also considered. In the case of the x-ray absorption spectroscopy (XAS) experiment, it is difficult to measure the part deeper than the penetration depth, where the x-ray intensity is reduced to $1/e$, because the contribution to the XAS decreases rapidly. On the other hand, in the case of elastic X-ray scattering, the contribution of a region

deeper than the penetration depth cannot be simply ignored because of the interference effect. For example, for the resonant energy (right panel), electric field intensities higher than 50% ($\sim 1/e$) of the incident beam are confined to a few or one or two PbTiO_3 layers close to the surface. Nevertheless, a total thickness fringe corresponding to eight bilayers can be observed in the specular reflectivity. Therefore, in soft x-ray scattering, unlike x-ray absorption spectroscopy, the maximum depth at which interference can occur may be deeper than the simple penetration depth."

REVIEWERS' COMMENTS

Reviewer #1 (Remarks to the Author):

I have reviewed the authors' response and the revised manuscript. The authors have adequately addressed my concerns and significantly improved the manuscript. My opinion is the work is now suitable for publication.

Reviewer #2 (Remarks to the Author):

The authors have addressed all referee concerns in the revised manuscript. Specifically, the treatment of absorption effects is now presented in more detail, which improves the technical soundness of the manuscript. I still think the authors have room to improve their message to the general Nature Communications readership, but I am glad to see significant improvements to the discussion of the importance and novelty of their research for the field of ferroelectric and multiferroic materials. I can now recommend this work for publication in Nature Communications.